# BAY 41-2272 Attenuates CTGF Expression via sGC/cGMP-Independent Pathway in TGFβ1-Activated Hepatic Stellate Cells

**DOI:** 10.3390/biomedicines8090330

**Published:** 2020-09-04

**Authors:** Po-Jen Chen, Liang-Mou Kuo, Yi-Hsiu Wu, Yu-Chia Chang, Kuei-Hung Lai, Tsong-Long Hwang

**Affiliations:** 1Department of Cosmetic Science, Providence University, Taichung 433719, Taiwan; litlep@hotmail.com; 2Department of General Surgery, Chang Gung Memorial Hospital at Chiayi, Chiayi 613016, Taiwan; kuo33410@yahoo.com.tw; 3Graduate Institute of Natural Products, College of Medicine, Chang Gung University, Taoyuan 333323, Taiwan; modemtw@gmail.com; 4Research Center for Chinese Herbal Medicine, Graduate Institute of Health Industry Technology, College of Human Ecology, Chang Gung University of Science and Technology, Taoyuan 333323, Taiwan; ycchang03@mail.cgust.edu.tw (Y.-C.C.); mos19880822@gmail.com (K.-H.L.); 5Graduate Institute of Pharmacognosy, College of Pharmacy, Taipei Medical University, Taipei 110301, Taiwan; 6Research Center for Food and Cosmetic Safety, College of Human Ecology, Chang Gung University of Science and Technology, Taoyuan 333323, Taiwan; 7Department of Anaesthesiology, Chang Gung Memorial Hospital, Taoyuan 333423, Taiwan; 8Department of Chemical Engineering, Ming Chi University of Technology, New Taipei City 243303, Taiwan

**Keywords:** Akt, BAY 41-2272, CTGF, hepatic stellate cell, sGC

## Abstract

Activation of hepatic stellate cells (HSCs) is a critical pathogenic feature of liver fibrosis and cirrhosis. BAY 41-2272 is a canonical non-nitric oxide (NO)-based soluble guanylyl cyclase (sGC) stimulator that triggers cyclic guanosine monophosphate (cGMP) signaling for attenuation of fibrotic disorders; however, the impact of BAY 41-2272 on HSC activation remains ill-defined. Transforming growth factor (TGF)β and its downstream connective tissue growth factor (CTGF or cellular communication network factor 2, CCN2) are critical fibrogenic cytokines for accelerating HSC activation. Here, we identified that BAY 41-2272 significantly inhibited the TGFβ1-induced mRNA and protein expression of CTGF in mouse primary HSCs. Indeed, BAY 41-2272 increased the sGC activity and cGMP levels that were potentiated by two NO donors and inhibited by a specific sGC inhibitor, ODQ. Surprisingly, the inhibitory effects of BAY 41-2272 on CTGF expression were independent of the sGC/cGMP pathway in TGFβ1-activated primary HSCs. BAY 41-2272 selectively restricted the TGFβ1-induced phosphorylation of Akt but not canonical Smad2/3 in primary HSCs. Together, we illustrate a unique framework of BAY 41-2272 for inhibiting TGFβ1-induced CTGF upregulation and HSC activation via a noncanonical Akt-dependent but sGC/cGMP-independent pathway.

## 1. Introduction

The overwhelming activation and proliferation of hepatic stellate cells (HSCs) is an important factor in hepatic fibrosis, an advanced pathogenic condition in liver cirrhosis, portal hypertension, and hepatocellular carcinoma. During acute or chronic injuries, star-like shaped HSCs shift from quiescent non-proliferative status to active state with contractile, proliferative, and fibrogenic properties, leading to pro-inflammatory growth factor generation and extracellular matrix (ECM) protein production to initiate matrix deposition in the liver [1,2,3]. Targeting activated HSCs has been proposed as a potential strategy to remedy liver diseases, returning activated HSCs to a quiescent status or repressing proliferation of activated HSCs. Therefore, identification of anti-fibrotic agents and their underlying molecular mechanisms of action is desirable for removing factors causing HSC activation [4,5,6,7].

The nitric oxide (NO)/soluble guanylate cyclase (sGC)/cyclic guanosine monophosphate (cGMP) pathway tightly controls important physiological functions in several human body organs; therefore, it has been considered as a worthwhile therapeutic target for many diseases including cardiopulmonary, neuronal, and fibrotic disorders [8,9,10,11]. sGC is a heterodimeric enzyme catalyzing the conversion of GTP to cGMP. The activation of native reduced sGC is canonically triggered by NO that in turn binds to the heme moiety of sGC to produce cGMP. There are various small molecules, sGC stimulators and sGC activators, that induce the activity of heme-dependent reduced sGC and heme-independent oxidized sGC, respectively [12,13]. Emerging evidence suggests that both sGC stimulators and activators affect tissue fibrosis inhibition via elevating intracellular levels of cGMP [11,14,15]; hence, pharmacological induction of sGC may be a promising approach to impede HSC activation.

Transforming growth factor (TGF)β is a pivotal fibrogenic cytokine to stimulate HSC activation. The TGFβ levels are typically low in quiescent HSCs but are upregulated and released from deposits in the ECM after liver injury. TGFβs activate HSCs for adaptation of fibroblast shape, contractility, proliferation, and migratory phenotype [16,17,18]. Targeting the local TGFβ activity or inhibiting TGFβ signaling is ideal for affecting HSC activation, which is the focus of this research strategy. The TGFβ-induced activation of HSCs during liver fibrosis takes place via canonical Smad-dependent or noncanonical Smad-independent signaling such as the Akt pathway [18,19]. It has been documented that TGFβ induced the connective-tissue growth factor (CTGF or cellular communication network factor 2, CCN2) expression via Smad and Stat3 signaling pathways in HSCs [20]. The pro-fibrotic CTGF is also upregulated and promotes the pathogenic process of liver fibrosis including cell proliferation, contractility, migration, and ECM production in activated HSCs [21,22,23]. The TGFβ–CTGF axis serves as a promising pathogenic pathway to affect HSC activation; however, its regulatory signaling is still elusive.

BAY 41-2272 is a regular non-nitric oxide (NO)-based sGC stimulator and is used to ameliorate the fibrosis in the lung, systemic sclerosis, peritoneal stripping, pulmonary hypertension, cardiomyocyte hypertrophy, and chronic renal disease [24,25,26,27,28,29]. To date, no studies have addressed the impact of BAY 41-2272 on HSC activation in detail. We investigated the possible capacity of BAY 41-2272 to affect sGC/cGMP in TGFβ1-activated mouse-primary HSCs. In the present study, BAY 41-2272 significantly attenuated the TGFβ1-induced CTGF expression and cell proliferation in primary HSCs. Unexpectedly, the inhibitory effects of BAY 41-2272 on CTGF upregulation were independent on the sGC/cGMP pathway. BAY 41-2272 alternatively inhibited the noncanonical Akt instead of canonical Smad2/3 pathway in TGFβ1-primed HSCs. Our findings provide a mechanistic basis for BAY 41-2272 activity as a potential agent for Akt-dependent inhibition of TGFβ1-instigated CTGF induction and HSC activation that is distinct from the canonical sGC/cGMP and Smad signaling.

## 2. Experimental Section

### 2.1. Reagents

BAY 41-2272 was purchased from Calbiochem (La Jolla, CA, USA). BAY 58-2667 was obtained from BioVision (Mountain, PA, USA). Recombinant human TGFβ1 was from R&D Systems (Minneapolis, MN, USA). TRIzol reagents were from Thermo Fisher Scientific (Waltham, MA, USA). iScript cDNA synthesis kit and power SYBR green PCR master mix were purchased from Bio-Rad (Hercules, CA, USA). WST-1 Assay Kit was obtained from Roche Applied Sciences (Mannheim, Germany). Antibodies against phospho-Akt (S473), phospho-Akt (T308), Akt, phospho-Smad2, phospho-Smad3, and Smad2/3 were purchased from Cell Signaling Technology (Beverly, MA, USA). Antibodies against GAPDH and CTGF were obtained from Santa Cruz Biotechnology (Santa Cruz, CA, USA). Other chemicals were purchased from Sigma (St. Louis, MO, USA).

### 2.2. Isolation of Mouse Primary HSCs

The study was approved by and followed the guidelines of the Institutional Animal Care and Use Committee of Chang Gung University, Taiwan (IACUC approval No.: CGU15-085 and date: 11 November 2015). We purchased 8- to 10-week-old C57BL/6 male mice from BioLASCO (Taiwan). Five mice were kept in a cage under a 12–12-h light–dark cycle and were provided with water and food. HSCs were isolated from livers of 8- to 10-week-old C57BL/6 male mice using a modified, previously described procedure [30,31]. Briefly, mice were euthanized by CO_2_ and perfused with normal saline via the inferior vena cava followed by injection with 0.1% (*w*/*v*) collagenase IV in HBSS. Livers were taken out and agitated in 0.1% (*w*/*v*) collagenase IV at 37 °C for 10 min. Cells were filtered through a nylon mesh and purified via Percoll gradient centrifugation. The isolated HSCs were cultured in Roswell Park Memorial Institute (RPMI) 1640 media supplemented with 10% fetal bovine serum and 10% horse serum at 37 °C in a humidified incubator with 5% CO_2_. The purity of HSCs was above 95%, as confirmed by their astrocytes, intracellular lipid droplets, and vitamin A autofluorescence [32]. HSCs were cultured for 7 days and then spread out as the next generation. The fifth to seventh generation of HSCs were seeded overnight for the following experiments.

### 2.3. Cell Viability

HSCs were cultured and starved in RPMI 1640 media for 24 h. Cells were treated with indicated compounds and then activated with TGFβ1 (5 ng/mL) for 0, 24, or 48 h. Cells were subsequently incubated with the WST-1 reagent at 37 °C for 2 h. The cell viability was monitored spectrophotometrically at 450 nm (Thermo Fisher Scientific; Waltham, MA, USA).

### 2.4. Western Blotting

Cell pellets were suspended in lysis buffer (50 mM 4-(2-hydroxyethyl)-1-piperazineethanesulfonic acid (HEPES), 100 mM NaCl, 1 mM ethylenediaminetetraacetic acid (EDTA), 2 mM Na_3_VO_4_, 5% 2-mercaptoethanol, and 1% Triton-X-100) and then centrifuged at 14,000× *g* for 20 min at 4 °C. Total protein concentration were determined by using a BCA Protein Assay Kit (Pierce, Rockford, IL, USA). We mixed 30 μg cell lysates with sample buffer (62.5 mM Tris-HCl (pH 6.8), 4% SDS, 5% β-mercaptoethanol, 2.5 mM Na_3_VO_4_, 0.0125% bromophenol blue, 10 mM di-N-pentyl phthalate, and 8.75% glycerol) at 100 °C for 5 min, separated by 10% or 12% sodium dodecyl sulfate polyacrylamide gel electrophoresis (SDS-PAGE), electrophoresed onto a nitrocellulose membrane, and assayed by immunoblotting with specific primary antibodies against phospho-Akt S473 (catalog no. 4060, Cell Signaling), phospho-Akt T308 (catalog no. 2965, Cell Signaling), Akt (catalog no. 4691, Cell Signaling), phospho-Smad2 (catalog no. 3108, Cell Signaling), phospho-Smad3 (catalog no. 9520, Cell Signaling), Smad2/3 (catalog no. 5678, Cell Signaling), GAPDH (catalog no. sc-32233, Santa Cruz Biotechnology), and CTGF (catalog no. sc-25440, Santa Cruz Biotechnology) at 4 °C for 16 h and horseradish peroxidase-conjugated secondary antibodies at room temperature for 1 h. The protein levels were determined using an enhanced chemiluminescence system and a densitometer (UVP, Upland, CA, USA).

### 2.5. Immunofluorescence Staining

Cells placed on cover slides were fixed with 4% formaldehyde for 10 min and then incubated with 5% goat serum for 60 min. Protein levels were determined using primary antibodies against CTGF (catalog no. sc-25440, Santa Cruz Biotechnology) in 5% bovine serum albumin (BSA) at room temperature for 1 h and fluorescein isothiocyanate (FITC)-conjugated secondary antibody for another 1 h. Nuclei were counterstained with Hoechst (1 μg/mL). Images were obtained by fluorescent microscopy (OLYMPUS IX 81; Olympus, Tokyo, Japan).

### 2.6. RNA Isolation and Quantitative Real-Time PCR

Total RNA was extracted from HSCs using TRIzol reagent and 1 µg RNA was used as a template for cDNA synthesis by iScript cDNA Synthesis Kit according to the manufacturer’s protocol. mRNA levels were determined using Power SYBR Green PCR Master Mix and quantitative PCR with CFX Connect Real-Time PCR Detection System (Bio-Rad, Hercules, CA, USA). Primers for mouse *CTGF* (5′-GGAATTGTGACCTGAGTGACT-3′ and 5′-TGAGCCAGCCATTTCTTAATAAAG-3′) and mouse *GAPDH* (5′-AAGGAGTAAGAAACCCTGGAC-3′ and 5′-GATGGAAATTGTGAGGGAGATG-3′) were used. The real-time PCR was conducted at 95 °C for 10 min, followed by 40 cycles of denaturation at 95 °C for 15 s, and annealing/extension at 60 °C for 1 min. PCR conditions were optimized to achieve a single peak by melting curve analysis on the CFX Connect system.

### 2.7. Determination of Intracellular cGMP Levels

HSCs were preincubated with ODQ (1 µM) and/or indicated phosphodiesterase (PDE) inhibitors for 15 min and then treated with BAY 41-2272 (0.3–10 µM), BAY 63-2521 (0.3–10 µM), or BAY 58-2667 (0.3–10 µM) in the presences or absence of S-nitroso-N-acetyl penicillamine (SNAP; 0.1 µM) or sodium nitroprusside (SNP; 0.1 µM) for another 15 min at 37 °C. The intracellular cGMP levels were detected using the commercial enzyme immunoassay (EIA) system (catalog no. RPN226, GE Healthcare, Little Chalfont, Buckinghamshire, UK) according to the manufacturer’s protocol.

### 2.8. sGC Activity Assay

HSCs were suspended and sonicated in sGC lysis buffer (25 mM Tris-HCl (pH 7.5), 250 mM sucrose, 2 mM EDTA, 5 mM MgCl_2_, 100 μM phenylmethylsulfonyl fluoride (PMSF), 10 μg/mL leupeptin, and 10 μM pepstain A). After centrifugation at 250× *g* for 5 min at 4 °C, the supernatant was centrifuged at 16,000× *g* for 15 min at 4 °C. The cytosolic fraction was mixed with the indicated test agents in sGC reaction buffer (25 mM Tris-HCl (pH 7.5), 15 mM MgCl_2_, 2 mM 3-isobutyl 1-methylxanthine, 15 mM creatine phosphate, 6 units creatine phosphokinase, 2 mM GTP) for 20 min at 30 °C and then stopped by heating at 100 °C for 3 min. cGMP levels were measured to determine the sGC activity using a commercial EIA system (catalog no. RPN226, GE Healthcare, Little Chalfont, Buckinghamshire, UK).

### 2.9. Statistical Analysis

Data were expressed as the mean ± standard error of mean (SD). Statistical comparisons were made between two groups using Student’s *t*-test. *p* < 0.05 was considered statistically significant.

## 3. Results

### 3.1. BAY 41-2272 sGC Stimulator Inhibited TGFβ1-Induced CTGF Expression and Cell Proliferation in Primary HSCs

To evaluate the biological significance of sGC/cGMP signaling in primary HSCs, we first evaluated the effect of a typical sGC stimulator BAY 41-2272 on TGFβ1-induced CTGF expression that is a critical pro-fibrotic cytokine for HSC activation. BAY 41-2272 dose-dependently (1–10 μM) inhibited the CTGF expression in primary TGFβ1-activated HSCs, as determined by Western blot and immunofluorescent staining (Figure 1A,B). The TGFβ1-induced proliferation of primary HSCs was also significantly restricted by BAY 41-2272 (10 μM; Figure 1C), suggesting that the compound may serve as an agent to ameliorate HSC activation.

BAY 41-2272 is a non-NO-based sGC stimulator; two distinctly different NO donors, SNP and SNAP, were used to synergistically enhance sGC/cGMP signaling. A selectively heme-site inhibitor of sGC, ODQ, was also used to examine the molecular actions of BAY 41-2272 in primary HSCs. BAY 41-2272 (3 or 10 μM) increased the intracellular cGMP levels in primary HSCs, and both SNP (0.1 μM) and SNAP (0.1 μM) synergistically enhanced the BAY 41-2272 (10 μM)-increased cGMP levels. ODQ blocked all BAY 41-2272 (10 μM)-induced cGMP production in primary HSCs (Figure 2A). Similarly, BAY 41-2272 (10 μM) induced the sGC activity in vitro that was further activated by SNP (0.1 μM) or SNAP (0.1 μM) and blocked by ODQ (1 μM) (Figure 2B), suggesting that BAY 41-2272 can trigger sGC/cGMP signaling in primary HSCs.

### 3.2. The BAY 41-2272-Inhibited CTGF Expression and Cell Proliferation Was not via sGC/cGMP Pathway in TGFβ1-Activated Primary HSCs

To examine whether the inhibitory effects of BAY 41-2272 on HSC activation is through the activated sGC/cGMP signaling, we checked the CTGF expression in TGFβ1-activated primary HSCs in the presence or absence of inhibitors, ODQ (sGC inhibitor), and KT5823 (cGMP-dependent protein kinase (PKG) inhibitor). Surprisingly, the BAY 41-2272-repressed mRNA and protein expressions of CTGF were not affected by ODQ (1 μM) or KT5823 (3 μM) in TGFβ1-activated HSCs (Figure 3). Moreover, the NO donors SNAP (0.1 μM) and SNP (0.1 μM) did not alter the inhibitory effects of BAY 41-2272 on the TGFβ1-induced mRNA and protein expression of CTGF in primary HSCs (Figure 4A–C). The BAY 41-2272-repressed cell proliferation was not changed in the presence of SNAP or SNP in TGFβ1-activated primary HSCs (Figure 4D), suggesting that the effects of BAY 41-2272 against HSC activation may be independent on its canonical sGC/cGMP signaling.

### 3.3. PDE9 Modulated the BAY 41-2272-Mediated sGC/cGMP Signaling But not CTGF Inhibition in Primary HSCs

cGMP-dependent phosphodiesterases (PDEs; PDE1, 2, 3, 5, 6, 9, 10, and 11) are responsible for the conversion from active cGMP to inactive 5′GMP [33]. To further understand the effect of the sGC/cGMP pathway on HSC activation, we used PDE inhibitors to examine the cGMP production and CTGF expression in primary HSCs. First, the non-specific PDE inhibitor 3-isobutyl-1-methylxanthine (IBMX) apparently increased the BAY 41-2272-induced intracellular cGMP levels in the presence or absence of SNAP or SNP. The cGMP levels were also inhibited by ODQ (Figure 5A). We also screened the isozyme-specific inhibitors of PDEs (vinpocetine for PDE1, erythro-9-(2-hydroxy-3-nonyl)adenine (EHNA) for PDE2, zaprinast and tadalafil for PDE5, BAY 73-6691 for PDE9) to check the BAY 41-2272-induced cGMP production in HSCs. The PDE9 inhibitor BAY 73-6691 showed the strongest effect on increasing the BAY 41-2272-induced cGMP levels (Figure 5B), suggesting that PDE9 is important for the sGC/cGMP pathway in HSCs. However, the PDE9 inhibitor did not change the inhibitory effect of BAY 41-2272 on CTGF expression in TGFβ1-activated HSCs (Figure 5C). This demonstrates further that the inhibitory effect of BAY 41-2272 on CTGF expression is via the sGC/cGMP-independent pathway.

### 3.4. The TGFβ1-Induced CTGF Expression is Independent of cGMP Formation in Primary HSCs

The sGC stimulator BAY 41-2272 has the potential of inhibiting HSC activation by prohibiting CTGF in TGFβ1-activated primary HSCs; however, this is separate from its ability to elicit sGC activity and cGMP formation. To further characterize the correlation between sGC/cGMP signaling and pro-fibrotic CTGF expression, we used another heme-dependent sGC stimulator, BAY 63-2521, and a heme-independent sGC activator, BAY 58-2667, to analyze the cGMP formation and TGFβ1-induced CTGF expression in primary HSCs. BAY 63-2521 (10 μM) alone increased the intracellular cGMP generation in HSCs that was further increased by synergetic incubation with NO donors SNP (0.1 μM) or SNAP (0.1 μM). Moreover, the PDE inhibitor IBMX increased the BAY 63-2521-induced cGMP levels in HSCs with or without SNAP or SNP. All the increased cGMP levels were inhibited by ODQ (1 μM) (Figure 6A). On the other hand, BAY 58-2667 had no effect on cGMP formation in HSCs, even in the presence of SNAP, SNP, IBMX, and/or ODQ (Figure 6B,C). These results indicate that heme-independent sGC is the major type of sGC in HSCs and may have unique functions in these cells.

BAY 63-2521 and BAY 58-2667 did not affect CTGF expression in TGFβ1-activated HSCs (Figure 7A,B). An analog of cGMP, 8-Br-cGMP (bromo-cGMP), also did not alter the TGFβ1-induced CTGF expression in HSCs (Figure 7C), suggesting that cGMP signaling may not be involved in CTGF production in TGFβ1-activated HSCs. Together, the activated sGC/cGMP signaling exhibits no advantage for blocking HSC activation via preventing CTGF expression.

### 3.5. BAY 41-2272 Selectively Inhibited the TGFβ1-Induced Akt Activation in Primary HSCs

The TGFβ-activated Smad and non-Smad pathways have been well-documented as the predominant fibrotic signaling in HSCs during liver fibrosis [16,17]. We checked the effect of BAY 41-2272 on Smad2/3 and Akt phosphorylation in TGFβ1-activated HSCs to investigate its downstream signaling. TGFβ1 (5 ng/mL) significantly triggered the phosphorylation of Smad2, Smad3, and Akt within 20 min, and BAY 41-2272 (10 μM) selectively reduced the TGFβ1-activated Akt but not Smad2/3 in primary HSCs (Figure 8A,B). The sGC inhibitor ODQ (1 μM) did not affect BAY 41-2272-inhibited Akt activation in TGFβ1-activated HSCs (Figure 8C), suggesting that the inhibitory effect of BAY 41-2272 may be mediated through Akt instead of the sGC/cGMP and Smad pathway in HSCs. Inhibition of phosphoinositide 3-kinase (PI3K) signaling in HSCs attenuates liver fibrosis [34,35]. Similarly, LY294002 PI3K inhibitor dose-dependently (1–10 μM) attenuated the Akt phosphorylation, CTGF expression, and cell viability in TGFβ1-activated primary HSCs (Figure 8D,E). During HSC activation, BAY 41-2272-based restriction of TGFβ1-induced CTGF expression may be selectively modulated through Akt-dependent but sGC/cGMP- and Smad-independent pathway.

## 4. Discussion

The fibrogenic cytokines TGFβ and CTGF trigger HSC activation as a pathogenic factor during liver fibrosis. Emerging evidence shows that stopping TGFβ-CTGF upregulation and sGC modulators are effective ways to control HSC activation [11,16,20,23]; however, the full understanding of their correlation is still elusive. BAY 41-2272 is a well-described sGC stimulator that is used to ameliorate fibrotic lesion [24,25,26,27,28,29], but its application in HSC remains undefined. In the present study, we manifested the pharmacological effects and mechanisms of BAY 41-2272 and sGC/cGMP signaling in TGFβ-activated primary mouse HSCs. BAY 41-2272 remarkably restricted the TGFβ1-induced CTGF upregulation and cell proliferation in mouse primary HSCs, along with increased cGMP levels and sGC activity. However, the inhibitory effects of BAY 41-2272 on TGFβ1-induced HSC activation and induction of sGC/cGMP signaling are astonishingly unconnected. BAY 41-2272 restricted the Akt activation instead of canonical Smad2/3 pathway in TGFβ1-activated HSCs, providing a plausible cellular basis for the mechanistic actions of BAY 41-2272 and Akt for HSC activation.

A growing body of evidence indicates that reversing to quiescent status or preventing proliferation is an effective way to treat activated HSCs [4,36,37]. TGFβ has been well-established for inducing HSCs into an activated status and becoming, in turn, myofibroblast-like cells with proliferative, contractile, and fibrogenic properties [16,17,18]. TGFβ signaling upregulates another fibrogenic factor, CTGF, which contributes to HSC activation [20,38,39]; inhibition of CTGF is an option for attenuating HSC activation [40,41,42]. In this study, we used isolated primary HSCs to address the possibility of targeting TGFβ1-induced CTGF expression. Both RNA and protein levels of CTGF are upregulated in TGFβ1-activated HSCs and hence the inhibitory effects of BAY 41-2272 on CTGF upregulation and HSC proliferation should be relevant to its anti-fibrotic effect on HSC activation (Figure 1 and Figure 3). Therefore, targeting CTGF upregulation may be an effective way of attenuating TGFβ-induced HSC activation.

The NO/sGC/cGMP pathway maintains various physiological homeostasis, and induction of sGC/cGMP signaling has been proposed to inhibit fibrogenesis [11,43]. However, the role of NO in regulating HSCs remains controversial. For example, the nitrovasodilator-mediated contraction and proliferation of HSCs are both NO/cGMP-dependent and -independent [14,44,45]. Lipopolysaccharide (LPS) triggers HSCs to release NO, but this is not related to HSC-induced hepatocyte proliferation [46,47]. Moreover, release of NO inhibits HSC activation [48]. Therefore, NO signaling may show diverse functions during HSC activation. SNP and SNAP act as NO donors via enzymatic oxidation and chemical reaction, respectively [49]. Both SNP and SNAP synergistically potentiated the BAY 41-2272-induced sGC activities and cGMP levels in HSCs (Figure 2); however, SNP and SNAP did not affect the BAY 41-2272-inhibited CTGF upregulation and proliferation of HSCs (Figure 4). We suggest that NO production may not participate in TGFβ1-induced HSC activation, although it augments sGC/cGMP signaling.

sGCs are expressed in HSCs and not in hepatocytes [50] and are divided into reduced and oxidized forms that possess heme-dependent and heme-independent properties, respectively [11,13]. Many sGC modulators have been developed: (1) sGC stimulators to sensitize the reduced and heme-containing sGC to NO, and (2) sGC activators to activate oxidized and heme-free sGC [51,52,53] to generate cGMP. Exposure of 8-Br-cGMP cGMP analog leads to an inhibition of TGFβ-induced fibrogenesis in renal, cardiac, and dermal fibroblasts [26,54,55], suggesting that the sGC/cGMP pathway may be preferential signaling for TGFβ-induced HSC activation. Here, the BAY 41-2272 sGC stimulator showed potent inducible and inhibitory effects on cGMP levels and TGFβ1-increased CTGF expression in HSCs (Figure 1 and Figure 2). However, only increased cGMP levels and sGC activity but not decreased CTGF expression was reversed by the ODQ sGC inhibitor in BAY 41-2272-treated HSCs (Figure 2 and Figure 3). Importantly, the elevation of cGMP signaling by KT5823 (PKG inhibitor), BAY 73-6691 (PDE9 inhibitor), and 8-Br-cGMP (cGMP analog) failed to alter the TGFβ1-induced CTGF expression in HSCs (Figure 3C, Figure 5C, and Figure 7C). Together, our results may support the existence of a unique framework making TGFβ-based HSC activation independent of sGC/cGMP signaling.

Emerging evidence has indicated the potential application of sGC modulators for treating a fibrotic lesion in the liver, skin, lung, and kidneys [11,15]. Riociguat (BAY 63-2521) is an sGC stimulator used for treating pulmonary hypertension that has attenuated cholestatic fibrogenesis and cirrhotic portal hypertension in rats [56,57]. IW-1973 and praliciguat sGC stimulators prevented hepatic fibrosis in models of non-alcoholic steatohepatitis [51,58]. Oral administration of BAY 60-2770 sGC activator prevented the carbon tetrachloride-induced hepatic fibrous collagen formation in rats [59]. Noticeably, BAY 41-2272 can quell fibrogenesis in various fibrotic disorders [24,25,26,27,28,29]; however, its effects on liver fibrosis is limited. In the present study, we evaluated the effects of BAY 41-2272 and BAY 63-2521 sGC stimulators, and BAY 58-2667 sGC activator on TGFβ1-mediated HSC activation. Only BAY 41-2272 inhibited the TGFβ1-induced CTGF expression in primary HSCs (Figure 1). BAY 63-2521 that synergistically increased cGMP levels with NO donors and BAY 58-2667 failed to alter the CTGF upregulation in TGFβ1-activated HSCs (Figure 6 and Figure 7). We propose that BAY 41-2272 may be distinct from other sGC modulators and exhibits two diverse pathways in HSCs: (1) canonical NO/sGC/cGMP-dependent signaling for undefined functions, and (2) sGC-independent repression of TGFβ1-triggered HSC activation.

In general, TGFβ binding with the TGFβ receptor leads to phosphorylation of the receptor-activated Smad2 and Smad3. Additionally, non-canonical Smad-independent signaling pathways such as PI3K/Akt are activated by TGFβ and provide a broad TGFβ-induced intracellular crosstalk [18,19]. Targeting PI3K/Akt signaling also restrains the progression of HSC activation, including cell proliferation and CTGF expression [34,35,60,61,62], supporting the biological significance of the PI3K/Akt pathway for TGFβ-dependent HSC activation. In this study, BAY 41-2272 reduced the TGFβ1-induced phosphorylation of Akt but not canonical Smad2/3. Pharmacological inhibition of PI3K/Akt also apparently blocked the TGFβ1-induced CTGF expression and proliferation of HSCs (Figure 8). Because the effect of BAY 41-2272 on Akt phosphorylation was transient, we cannot exclude the possibility that the inhibitory effects of BAY 41-2272 in HSCs are mediated by other targets. This is the first time that BAY 41-2272 was explored to treat TGFβ-mediated HSC activation via targeting the PI3K/Akt that plays an important role in this process.

## 5. Conclusions

We illustrate that the BAY 41-2272 sGC stimulator attenuates the TGFβ1-induced CTGF expression and cell proliferation through the Akt signaling and not the sGC/cGMP pathway in mouse primary HSCs. Our results provide a novel insight into the TGFβ1- and BAY 41-2272-based regulatory networks and molecular profiles for HSC activation.

## Figures and Tables

**Figure 1 biomedicines-08-00330-f001:**
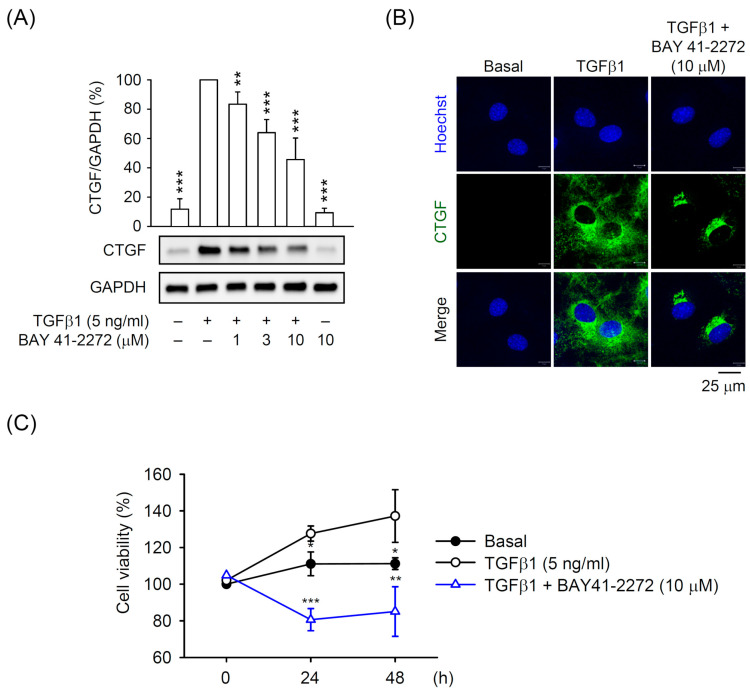
BAY 41-2272 repressed the transforming growth factor (TGF)β1-induced connective tissue growth factor (CTGF) expression in hepatic stellate cells (HSCs). HSCs were preincubated with BAY 41-2272 (1–10 µM) for 30 min before activation by TGFβ1 (5 ng/mL) for another (**A**,**B**) 6 h or (**C**) 24–48 h. Expression of CTGF and GAPDH was determined by (**A**) Western blot and (**B**) immunofluorescent staining using the corresponding antibodies. (**C**) Cell viability was measured using the WST-1 Assay Kit and monitored spectrophotometrically at 450 nm. All data are expressed as mean ± SD (*n* = 3). * *p* < 0.05, ** *p* < 0.01, *** *p* < 0.001 compared with the TGFβ1 alone.

**Figure 2 biomedicines-08-00330-f002:**
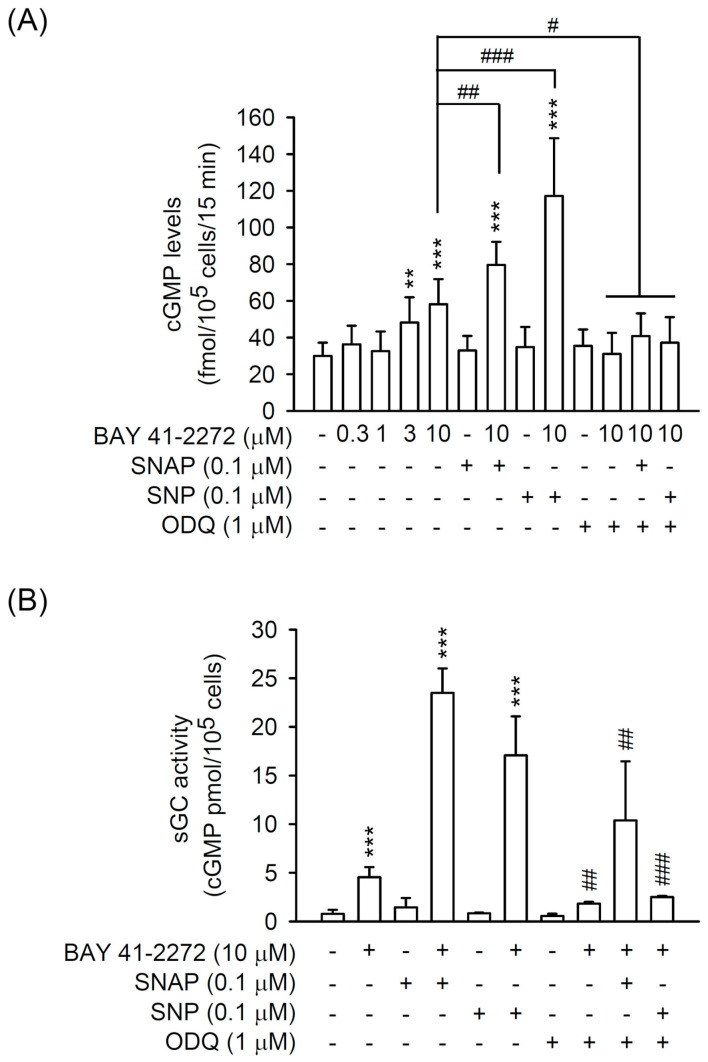
BAY 41-2272 increased the soluble guanylyl cyclase (sGC) activity and cyclic guanosine monophosphate (cGMP) formation in HSCs. **(A)** HSCs were preincubated with or without ODQ (1 µM) sGC inhibitor for 15 min and then treated with BAY 41-2272 (0.3–10 µM) in the presence or absence of S-nitroso-N-acetyl penicillamine (SNAP) (0.1 µM) or sodium nitroprusside (SNP) (0.1 µM) NO donors for another 15 min. (**B**) The cytosolic fractions from HSCs were mixed with or without ODQ (1 µM), BAY 41-2272 (10 µM), SNAP (0.1 µM), and/or SNP (0.1 µM) at 30 °C for 20 min. The cGMP levels were assayed using enzyme immunoassay (EIA) kits. All data are expressed as mean ± SD (*n* = 4). ** *p* < 0.01, *** *p* < 0.001 compared with the basal; ^#^
*p* < 0.05, ^##^
*p* < 0.01, ^###^
*p* < 0.001 compared with (**A**) BAY 41-2272 (10 µM) alone and (**B**) the corresponding BAY 41-2272 with or without SNAP or SNP.

**Figure 3 biomedicines-08-00330-f003:**
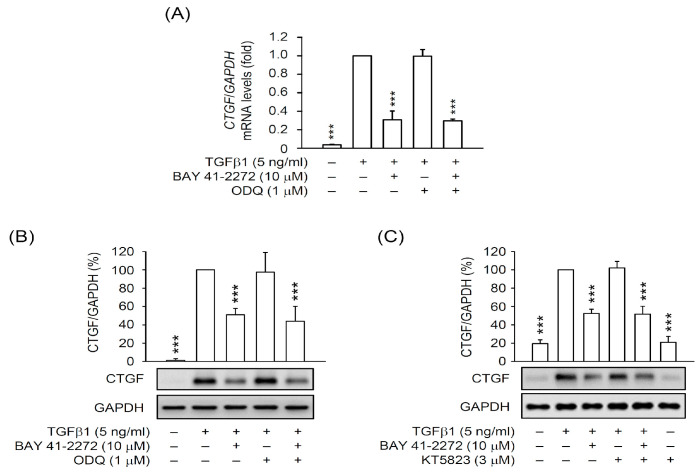
The inhibitory effect of BAY 41-2272 on CTGF expression was independent of the sGC pathway in TGFβ1-activated HSCs. HSCs were pretreated with or without ODQ (1 µM) or KT5823 (cGMP-dependent protein kinase (PKG) inhibitor) (3 µM) PKG inhibitors for 15 min. HSCs were sequentially incubated with BAY 41-2272 (10 µM) for 30 min before activation by TGFβ1 (5 ng/mL) for another 6 h. **(A)** mRNA levels of *CTGF* or *GAPDH* were determined by quantitative RT-PCR. (**B**,**C**) Expressed CTGF and GAPDH proteins were determined by Western blot using the corresponding antibodies. All data are expressed as mean ± SD (*n* = 3). *** *p* < 0.001 compared with the TGFβ1 alone.

**Figure 4 biomedicines-08-00330-f004:**
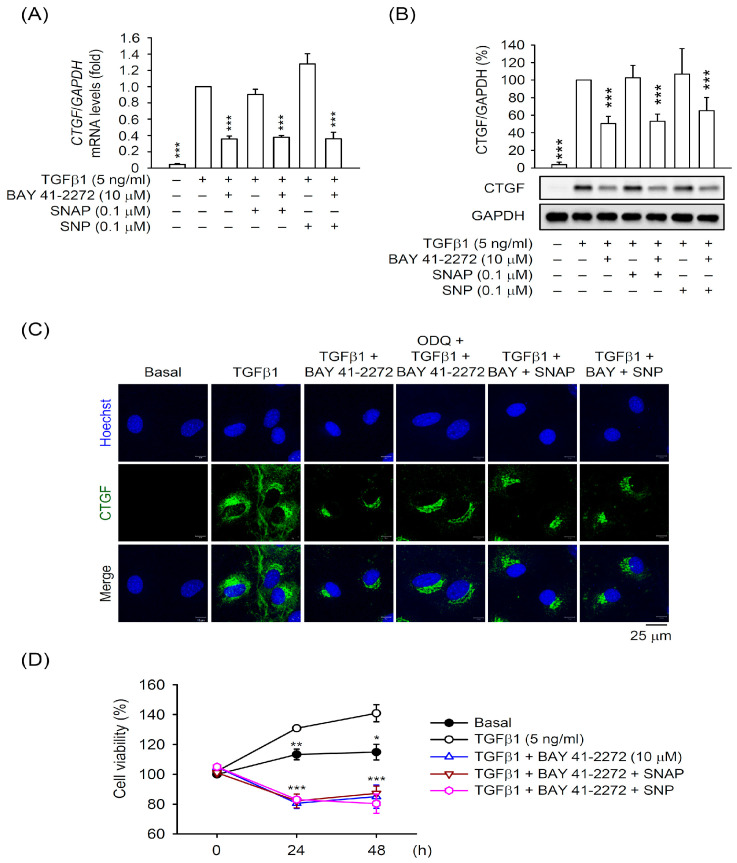
NO donors did not affect the inhibition of TGFβ1-activated HSCs by BAY 41-2272. HSCs were preincubated with BAY 41-2272 (10 µM) in the presence or absence of SNAP (0.1 µM) or SNP (0.1 µM) for 30 min before activation by TGFβ1 (5 ng/mL) for another (**A**–**C**) 6 h or (**D**) 24–48 h. (**A**) mRNA levels of CTGF and GAPDH were determined by quantitative RT-PCR. (**B**,**C**) Expressed CTGF and GAPDH proteins were measured by (**B**) Western blot and (**C**) immunofluorescent staining using the corresponding antibodies. (**D**) Cell viability was measured using the WST-1 assay and monitored spectrophotometrically at 450 nm. All data are expressed as mean ± SD (*n* = 3). * *p* < 0.05, ** *p* < 0.01, *** *p* < 0.001 compared with the TGFβ1 alone.

**Figure 5 biomedicines-08-00330-f005:**
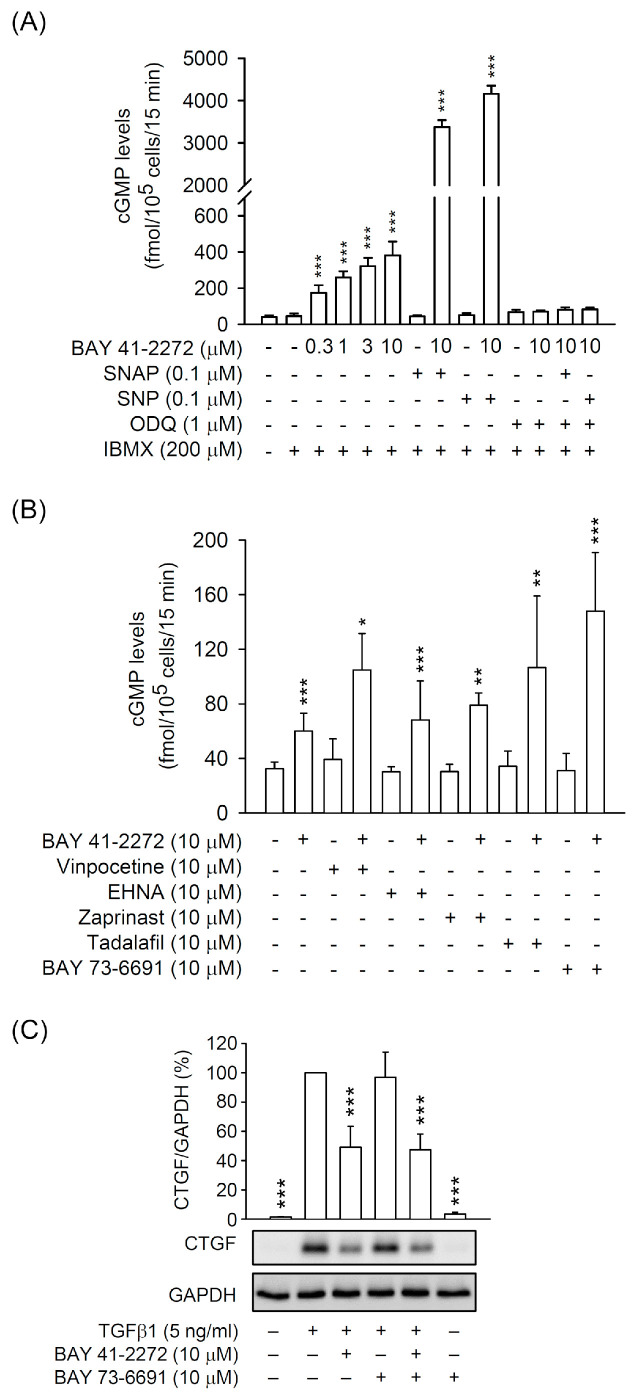
Phosphodiesterase (PDE) inhibitors increase the BAY 41-2272-mediated cGMP formation but not CTGF expression in HSCs. (**A**,**B**) HSCs were preincubated with or without (**A**) IBMX (200 µM) and/or ODQ (1 µM) or (**B**) specific PDE inhibitors for 15 min and then treated with BAY 41-2272 (0.3–10 µM) in the presence or absence of SNAP (0.1 µM) or SNP (0.1 µM) for another 15 min. The cGMP levels were assayed using EIA kits. (**C**) HSCs were pretreated with or without BAY 73-6691 (10 µM) PDE9 inhibitor for 15 min. HSCs were sequentially incubated with BAY 41-2272 (10 µM) for 30 min before activation by TGFβ1 (5 ng/mL) for another 6 h. Expressed CTGF and GAPDH were measured by Western blot using the corresponding antibodies. All data are expressed as mean ± SD (*n* = 4). * *p* < 0.05, ** *p* < 0.01, *** *p* < 0.001 compared with the basal (**A**,**B**) or TGFβ1 alone (**C**).

**Figure 6 biomedicines-08-00330-f006:**
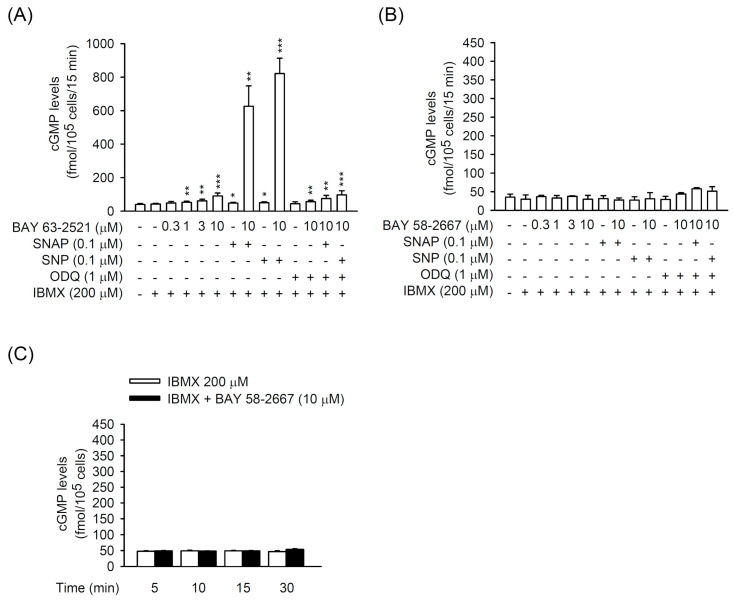
Effects of BAY 63-2521 and BAY 58-2667 on cGMP formation in HSCs. HSCs were preincubated with or without IBMX (200 µM) and/or ODQ (1 µM) for 15 min and then treated with (**A**) BAY 63-2521 (0.3-10 µM) or (**B**,**C**) BAY 58-2667 (0.3–10 µM) in the presence or absence of SNAP (0.1 µM) or SNP (0.1 µM) for another (**A**,**B**) 15 min or (**C**) various time intervals (5–30 min). The cGMP levels were assayed using EIA kits. All data are expressed as mean ± SD (*n* = 4). * *p* < 0.05, ** *p* < 0.01, *** *p* < 0.001 compared with the basal.

**Figure 7 biomedicines-08-00330-f007:**
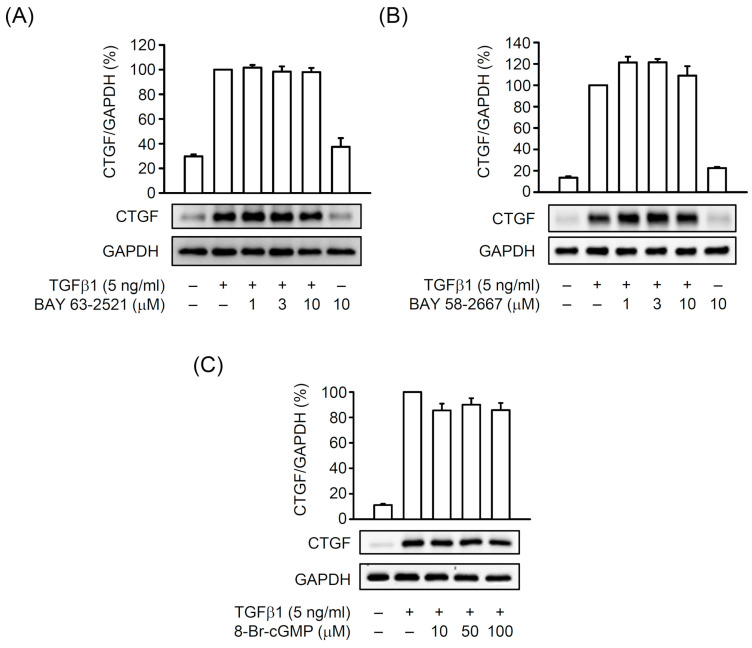
Effects of BAY 63-2521, BAY 58-2667, and 8-Br-cGMP on TGFβ1-induced CTGF expression in HSCs. HSCs were preincubated with (**A**) BAY 63-2521 (1–10 µM), (**B**) BAY 58-2667 (1–10 µM), or (**C**) 8-Br-cGMP (10–100 µM) for 30 min before activation by TGFβ1 (5 ng/mL) for another 6 h. Expressed CTGF and GAPDH were determined by Western blot using the corresponding antibodies. All data are expressed as mean ± SD (*n* = 3).

**Figure 8 biomedicines-08-00330-f008:**
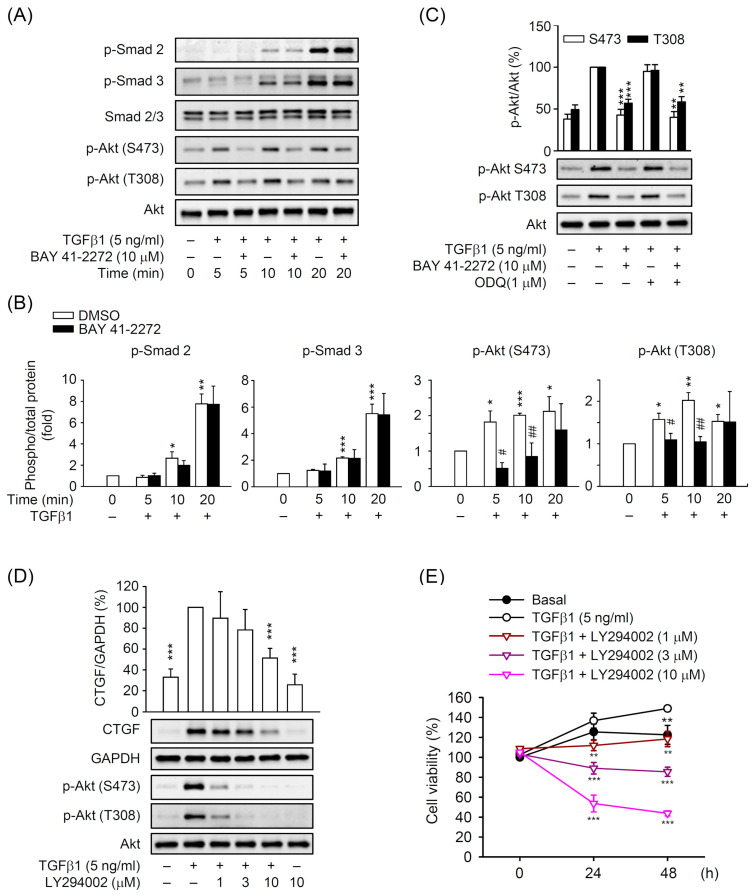
The BAY 41-2272-inhibited CTGF expression was dependent on Akt signaling in TGFβ1-activated HSCs. HSCs were pretreated with or without ODQ (1 µM) for 15 min. HSCs were sequentially incubated with (**A**–**C**) BAY 41-2272 (10 µM) or (**D**,**E**) LY294002 (1–10 µM) Akt signaling inhibitor for 30 min before activation by TGFβ1 (5 ng/mL) for another (**A**,**B**) 5–20 min, (**C**,**D**) 10 min, or (**E**) 24–48 h. (**A**–**D**) Protein levels were determined by Western blot using antibodies against p-Smad2, p-Smad3, Smad2/3, p-Akt (S473 or T308), Akt, CTGF, or GAPDH. (**E**) Cell viability was measured using the WST-1 assay and monitored spectrophotometrically at 450 nm. All data are expressed as mean ± SD (*n* = 3). * *p* < 0.05, ** *p* < 0.01, *** *p* < 0.001 compared with the basal (**B**) or TGFβ1 alone (**C**–**E**). ^#^
*p* < 0.05, ^##^
*p* < 0.01 compared with the corresponding DMSO group.

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
