# Peer review of "BAY 41-2272 Attenuates CTGF Expression via sGC/cGMP-Independent Pathway in TGFβ1-Activated Hepatic Stellate Cells"

_biomedicines, 2020, doi:10.3390/biomedicines8090330_

Round 1

Reviewer 1 Report

The manuscript by Chen et al., describes the function of BAY 41-2272 as a potential agent for Akt-dependent inhibition of TGFβ1-caused connective tissue growth factor CTGF induction and activation of hepatic stellate cells that is distinct from the canonical sGC/cGMP and Smad signaling.

Overall, the work is valuable and of major interest. The results obtained are intriguing, the experiments are nicely explained, and the manuscript is well written and logical in its presentation and style. In my opinion this paper deserves to be published in the pages of the Biomedicines.

I have just 3 minor remarks:

The lines numeration is completely wrong

Pag.3 line 1-4: You should show the “product code” of all the antibodies used in this work. For instance, which anti-AKT did you use? Cell Signaling produces and sells at least half a dozen anti-AKT. This information is very important for the reader and for the reproducibility of your experiments.

Pag.3 lines 1-4: In my opinion, it is more useful moving the antibodies list from “Reagents” to “Western Blotting” and “Immunofluorescence staining”.

Author Response

Reviewer #1

Comments and Suggestions for Authors

The manuscript by Chen et al., describes the function of BAY 41-2272 as a potential agent for Akt-dependent inhibition of TGFβ1-caused connective tissue growth factor CTGF induction and activation of hepatic stellate cells that is distinct from the canonical sGC/cGMP and Smad signaling.

Overall, the work is valuable and of major interest. The results obtained are intriguing, the experiments are nicely explained, and the manuscript is well written and logical in its presentation and style. In my opinion this paper deserves to be published in the pages of the Biomedicines.

 I have just 3 minor remarks:

1. The lines numeration is completely wrong

Reply: Thank you for your comments. The lines numeration has been checked according to the Template file from Biomedicines web site.

2. Pag.3 line 1-4: You should show the “product code” of all the antibodies used in this work. For instance, which anti-AKT did you use? Cell Signaling produces and sells at least half a dozen anti-AKT. This information is very important for the reader and for the reproducibility of your experiments.

Reply: Thank you for your suggestions. The information of all antibodies have been added in the revised manuscript and amended accordingly.

Page 3, 2.4. Western Blotting:

immunoblotting with specific primary antibodies against phospho-Akt S473 (catalog no. 4060, Cell Signaling), phospho-Akt T308 (catalog no. 2965, Cell Signaling), Akt (catalog no. 4691, Cell Signaling), phospho-Smad2 (catalog no. 3108, Cell Signaling), phospho-Smad3 (catalog no. 9520, Cell Signaling), Smad2/3 (catalog no. 5678, Cell Signaling), GAPDH (catalog no. sc-32233, Santa Cruz Biotechnology), and CTGF (catalog no. sc-25440, Santa Cruz Biotechnology) at 4 °C for 16 h and horseradish peroxidase-conjugated secondary antibodies at room temperature for 1 h.

Page 4, 2.5. Immunofluorescence staining:

Protein levels were determined using primary antibodies against CTGF (catalog no. sc-25440, Santa Cruz Biotechnology) in 5% BSA at room temperature for 1 h and FITC-conjugated secondary antibody for another 1 h.

3. Pag.3 lines 1-4: In my opinion, it is more useful moving the antibodies list from “Reagents” to “Western Blotting” and “Immunofluorescence staining”.

Reply: Thank you for your suggestions. The information of antibodies have been added in the revised manuscript as described in Point 2.

Reviewer 2 Report

This papers describes the effect of the cGMP-raising and putative anti-fibrotic drug BAY 41-2272 on the expression of CTGF in activated mouse hepatic stellate cells (HSCs) in vitro and on proliferation of these cells. The experiments seem well executeed and corroborate an anti-fibrotic effect of BAY 41-2272 on HSC as it inhibits the expression of CTGF and proliferation. The experiments also show that the effect of the drug is not mediated by cGMP but do not indicate the target of the drug.

Major points

The authors can not exclude that the effects of BAY 41-2272 in mouse HSCs are mediated by other targets than Akt-2 as the effects on Akt-2 phosphorylation seem transient and are moderate. Therefore they should omit the term “via Akt-2 dependent “ from the title and discuss other possibilities in the discussion.

Author Response

Reviewer #2

Comments and Suggestions for Authors

This papers describes the effect of the cGMP-raising and putative anti-fibrotic drug BAY 41-2272 on the expression of CTGF in activated mouse hepatic stellate cells (HSCs) in vitro and on proliferation of these cells. The experiments seem well executeed and corroborate an anti-fibrotic effect of BAY 41-2272 on HSC as it inhibits the expression of CTGF and proliferation. The experiments also show that the effect of the drug is not mediated by cGMP but do not indicate the target of the drug.

Major points

The authors can not exclude that the effects of BAY 41-2272 in mouse HSCs are mediated by other targets than Akt-2 as the effects on Akt-2 phosphorylation seem transient and are moderate. Therefore they should omit the term “via Akt-2 dependent “ from the title and discuss other possibilities in the discussion.

Reply: Thank you for your suggestions. We have amended the revised manuscript accordingly.

Page 1, Title:

BAY 41-2272 attenuates CTGF expression via sGC/cGMP-independent pathway in TGFβ1-activated hepatic stellate cells

Discussion, Page 14, Line 10:

BAY 41-2272 reduced the TGFβ1-induced phosphorylation of Akt but not canonical Smad2/3. Pharmacological inhibition of PI3K/Akt also apparently blocked the TGFβ1-induced CTGF expression and proliferation of HSCs (Figure 8). Because the effect of BAY 41-2272 on Akt phosphorylation was transient, we cannot exclude the possibility that the inhibitory effects of BAY 41-2272 in HSCs are mediated by other targets.

Reviewer 3 Report

The proposed manuscript by Chen et al. presents the results from a study aimed to investigate the impact of the canonical non-nitric oxide (NO)-based soluble guanylyl cyclase (sGC) stimulator BAY 41-2272 on hepatic stellate cells (HSCs) isolated from C57Bl/6 mice. Results demonstrated that BAY 41-2272 significantly downregulates TGFbeta1-induced expression of connective tissue growth factor (CTGF) and upregulated the activity of sGC and cyclic guanosine monophosphate (cGMP) levels. These effects of BAY 41-2272 were dependent on Akt but not on the sGC/cGMP signaling pathway. Based on the obtained findings the authors of the study concluded that the present results provide a novel insigt into the TGFbeta1- and BAY 41 2272-based regulatory network and molecular profiles for HSC activation.

In my opinion, this is a well designed and performed study. The manuscript is written very well and I do not find any significant incorrectness. My following comments are of minor character.

Specific comments and recommendations:

- Experimental Section, Subsection 2.2.: Please provide details on where the mice were obtained from and, the living conditions and how were they euthanized.

- Experimental Section, Subsection 2.4.: The “specific antibodies” need to be listed (including their catalog numbers) as well as the incubation time should be provided. What was the total protein concentration to which the samples were notmalized?

- Experimental Section, Subsection 2.5.: Similarly to my previous comment, the authors need to provide details on the antibodies used.

- Experimental Section, Subsection 2.6.: What total RNA concentration was used for the RT reaction? Please provide details on the qPCR conditions.

- Experimental Section, Subsection 2.8.: Please provide the catalog number of the sGC activity EIA kit used.

Author Response

Reviewer #3

Comments and Suggestions for Authors

The proposed manuscript by Chen et al. presents the results from a study aimed to investigate the impact of the canonical non-nitric oxide (NO)-based soluble guanylyl cyclase (sGC) stimulator BAY 41-2272 on hepatic stellate cells (HSCs) isolated from C57Bl/6 mice. Results demonstrated that BAY 41-2272 significantly downregulates TGFbeta1-induced expression of connective tissue growth factor (CTGF) and upregulated the activity of sGC and cyclic guanosine monophosphate (cGMP) levels. These effects of BAY 41-2272 were dependent on Akt but not on the sGC/cGMP signaling pathway. Based on the obtained findings the authors of the study concluded that the present results provide a novel insigt into the TGFbeta1- and BAY 41 2272-based regulatory network and molecular profiles for HSC activation.

In my opinion, this is a well designed and performed study. The manuscript is written very well and I do not find any significant incorrectness. My following comments are of minor character.

Specific comments and recommendations:

1. Experimental Section, Subsection 2.2.: Please provide details on where the mice were obtained from and, the living conditions and how were they euthanized.

Reply: Thank you for your suggestions. The revised manuscript has been amended accordingly.

Page 3, 2.2. Isolation of mouse primary HSCs:

8- to 10-weeks old C57BL/6 male mice were purchased from BioLASCO (Taiwan). Five mice were kept in a cage under a 12–12-h light–dark cycle and provided with water and food. HSCs were isolated from livers of C57BL/6 male mice using a modified, previously described procedure [30,31]. Briefly, mice were euthanized by CO2 and perfused with normal saline

2. Experimental Section, Subsection 2.4.: The “specific antibodies” need to be listed (including their catalog numbers) as well as the incubation time should be provided. What was the total protein concentration to which the samples were notmalized?

Reply: Thank you for your suggestions. The information of antibodies (including catalog numbers and incubation time) and total protein concentration have been added in the revised manuscript accordingly.

Page 3, 2.4. Western Blotting:

Line 27:  Total protein concentration were determined by using a BCA protein assay kit (Pierce, Rockford, IL, USA). 30 μg cell lysates were mixed with sample buffer

Line 32: immunoblotting with specific primary antibodies against phospho-Akt S473 (catalog no. 4060, Cell Signaling), phospho-Akt T308 (catalog no. 2965, Cell Signaling), Akt (catalog no. 4691, Cell Signaling), phospho-Smad2 (catalog no. 3108, Cell Signaling), phospho-Smad3 (catalog no. 9520, Cell Signaling), Smad2/3 (catalog no. 5678, Cell Signaling), GAPDH (catalog no. sc-32233, Santa Cruz Biotechnology), and CTGF (catalog no. sc-25440, Santa Cruz Biotechnology) at 4 °C for 16 h and horseradish peroxidase-conjugated secondary antibodies at room temperature for 1 h.

3. Experimental Section, Subsection 2.5.: Similarly to my previous comment, the authors need to provide details on the antibodies used.

Reply: Reply: Thank you for your suggestions. The information of antibodies have been added in the text accordingly.

Page 4, 2.5. Immunofluorescence staining:

Protein levels were determined using primary antibodies against CTGF (catalog no. sc-25440, Santa Cruz Biotechnology) in 5% BSA at room temperature for 1 h and FITC-conjugated secondary antibody for another 1 h.

4. Experimental Section, Subsection 2.6.: What total RNA concentration was used for the RT reaction? Please provide details on the qPCR conditions.

Reply: Thank you for your suggestions. The text has been amended accordingly.

Page 4, 2.6. RNA isolation and quantitative real-time PCR

Total RNA were extracted from HSCs using TRIzol reagent and 1 µg RNA were used as a template for cDNA synthesis by iScript™ cDNA Synthesis Kit according to the manufacturer’s protocol. mRNA levels were determined using Power SYBR Green PCR Master Mix and quantitative PCR with CFX Connect real-time system (Bio-Rad, Hercules, CA, USA). Primers for mouse CTGF (5’-GGAATTGTGACCTGAGTGACT-3’ and 5’-TGAGCCAGCCATTTCTTAATAAAG-3’) and mouse GAPDH (5’-AAGGAGTAAGAAACCCTGGAC-3’ and 5’-GATGGAAATTGTGAGGGAGATG-3’) were used. The real-time PCR was conducted at 95 °C for 10 min followed by 40 cycles of denaturation at 95 °C for 15 sec, annealing/extension at 60 °C for 1 min. PCR conditions were optimized to achieve a single peak by melting curve analysis on CFX Connect system.

5. Experimental Section, Subsection 2.8.: Please provide the catalog number of the sGC activity EIA kit used.

Reply: Thank you for your suggestions. The catalog number of commercial EIA system have been added in the revised manuscript.

Page 4, 2.8. sGC activity assay

cGMP levels were measured to determine the sGC activity using a commercial EIA system (catalog no. RPN226, GE Healthcare, Little Chalfont, Buckinghamshire, UK).